# VaccineDesigner: A Web-Based Tool for Streamlined Multi-Epitope Vaccine Design

**DOI:** 10.3390/biology14081019

**Published:** 2025-08-07

**Authors:** Dimitrios Trygoniaris, Anna Korda, Anastasia Paraskeva, Esmeralda Dushku, Georgios Tzimagiorgis, Minas Yiangou, Charalampos Kotzamanidis, Andigoni Malousi

**Affiliations:** 1Lab of Biological Chemistry, School of Medicine, Aristotle University of Thessaloniki, 54124 Thessaloniki, Greece; tzimagio@auth.gr; 2Department of Genetics, Development & Molecular Biology, School of Biology, Aristotle University of Thessaloniki, 54124 Thessaloniki, Greece; korda.anna01@gmail.com (A.K.); paraskevaa@bio.auth.gr (A.P.); yiangou@bio.auth.gr (M.Y.); 3Veterinary Research Institute of Thessaloniki, Campus of Thermi, 57001 Thermi, Greece; nesmeral@bio.auth.gr (E.D.); kotzam@elgo.gr (C.K.); 4Genomics and Epigenomics Translational Research Group, Center for Interdisciplinary Research and Innovation, 57001 Thessaloniki, Greece

**Keywords:** multi-epitope vaccines, reverse vaccinology, immunoinformatics, pathogen genomes, B cell, CTL, HTL epitope prediction, web-based tool

## Abstract

Designing safe and effective vaccines is a major challenge in protecting people and animals from infectious diseases. Traditional vaccine development can be slow, expensive, and sometimes risky. In recent years, scientists have turned to computer-based methods to speed up the process by identifying small parts of a virus or bacteria, called epitopes, that can trigger the body’s immune system to fight off infections. However, using many different computer tools separately can be difficult and time-consuming. To solve this problem, we developed VaccineDesigner, a free and easy-to-use platform that brings together all the steps needed to design modern vaccines in one place. The platform helps researchers choose the best epitopes for stimulating strong and safe immune responses, combine them into new vaccine candidates, and test them using built-in evaluation tools. We showed that VaccineDesigner works well by applying it to disease-causing bacteria and viruses, and we confirmed that predictions match real-world experimental data. VaccineDesigner makes vaccine development a faster and more reliable process. By making advanced technology accessible through a simple online interface, VaccineDesigner can help researchers create better vaccines to prevent disease and protect public health more efficiently.

## 1. Introduction

Developing effective and safe vaccines against pathogens is a leading research priority in immunology and public health. Vaccine development is typically based on live attenuated pathogens, inactivated agents, subunit formulations, or recombinant protein antigens. However, these methods raise several safety concerns, struggle with pathogen cultivation, are time-consuming, and, therefore, are impractical particularly during pandemics [1]. Over the past two decades, computational methods have become essential tools for enhancing efficacy, safety, and population coverage, rationalizing vaccine design. Reverse vaccinology (RV) [2] is fundamentally based on using computational methods to sift through the genetic information of a pathogen and identify efficient vaccine candidates that engage with the immune system. However, the identification of suitable proteins in pathogen proteomes is only the first step in computational vaccine design. Novel approaches have revolutionized our ability to predict and strategically harness epitopes [3], i.e., tiny peptide fragments of antigens capable of inducing protective immunity against pathogens. Since epitope-based approaches have been systematically used for some years now [4], a reasonable advancement would be the identification of immunodominant epitopes in vaccine candidates and their combination in efficient multi-epitope peptide constructs.

Multi-epitope vaccines constitute a preferable strategy for several reasons, including increased specificity over traditional methods, customizability, stability, and convenient production [5]. In addition, multi-epitope vaccines can include different types of epitopes, epitopes of different antigens, epitopes of different pathogens for cross-protection, or epitopes of different protein isoforms for broader population coverage. On the other hand, multi-epitope vaccines are often inadequately immunogenic, and optimization is difficult as it is not feasible to exhaustively examine all possible peptide combinations [5]. Current methods for multi-epitope peptide construction are known from fusion protein technology [6], and the strategic design of multi-epitope vaccines offers a promising approach for addressing diseases against a wide spectrum of pathogens, including parasites (e.g., visceral leishmaniasis [7] and onchocerciasis [8]), viruses (e.g., COVID-19 [9,10] and Ebola [11]), and bacterial infections (e.g., Helicobacter pylori [12] and Klebsiella pneumoniae [13]).

The components of a multi-epitope vaccine include B-cell epitopes for humoral responses (B cells), cytotoxic T lymphocyte epitopes for cellular immunity (CTLs), and helper T lymphocyte epitopes (HTLs) for immune regulation [14]. B-cell epitopes consist of surface-accessible clusters of amino acids and activate immune responses through antibody production [15]. T-cell epitopes consist of small peptide fragments and activate immune responses by presentation to antigen-presenting cells (APCs) through MHC class I or class II molecules. Cytotoxic T cells respond to MHC class I-restricted peptides, called CTL epitopes, whereas helper T cells target MHC class II-restricted peptides, referred to as HTL epitopes [16]. A multi-epitope vaccine has the unique ability to simultaneously activate both humoral and cellular immunity while avoiding the presence of elements that could provoke harmful reactions [17]. In the final multi-epitope construct, it is important to include components with adjuvant capacity to maximize immunogenicity as well as suitable linkers/spacers between epitopes for optimal proteasome processing and minimizing of junctional epitopes [18].

Over the past decades, epitope-based computational approaches have revolutionized vaccine design. BepiPred 3.0 [19], BCEPRED [20], and ABCpred [21] rely on machine learning methods to predict linear B-cell epitopes, while CBTOPE [22] integrates structural data to predict conformational epitopes. DiscoTope 3.0 [23] and ElliPro [24] focus on the prediction of discontinuous B-cell epitopes based on the spatial arrangements of amino acids. To address T-cell epitope prediction, NetMHCpan-4.0 [25] and NetMHCIIpan-4.0 [25] quantify peptide MHC binding affinities using a neural network architecture, while PROPRED [26] is primarily focused on identifying promiscuous MHC class-II binding regions. A wide range of tools have been developed to evaluate the properties of the predicted epitopes. ProtParam [27], VaxiJen [28], and Vaxign-ML [29] assess the physicochemical characteristics and the antigenic, protegenicity potential of the predicted epitopes, respectively. IFNepitope [30] predicts IFN-γ-inducing epitopes, while AlgPred 2.0 [31] and ToxinPred2 [32] assess allergenic and toxic potential. Novel approaches attempt to combine the outputs of multiple complementary tools in order to provide more effective and generalizable solutions. Vaxign2 [33], iVax [34], and Vacceed [35] are examples of workflows facilitating the discovery of vaccine targets in bacterial and eukaryotic pathogens. In support of these analyses, the Immune Epitope Database and Analysis Resource (IEDB-AR) [36] provides a robust and valuable repository and analysis tool for experimentally validated immune epitopes and predictions.

Despite the advances in building integrated software tools [37], the orchestration of RV resources remains technically challenging [38] due to their structural heterogeneities and lack of interoperability [39]. Furthermore, existing frameworks have functional and technical limitations regarding their ability to address epitope prediction for all cell types, the ability to construct multi-epitope vaccines, and their ease of use.

To address these limitations, we introduce VaccineDesigner, a novel, open-access, web-based platform that provides a fully integrated, end-to-end workflow for rational multi-epitope vaccine design. The platform supports comprehensive epitope prediction for both B-cell and T-cell responses, facilitating vaccine development for a broad range of human and non-human pathogens. Rather than requiring manual integration of heterogeneous and distributed tools, VaccineDesigner executes cascading tasks seamlessly within a unified, customizable graphical interface.

Key functionalities include multi-epitope vaccine generation, candidate prioritization, population coverage estimation, molecular mimicry analysis, and proteasome cleavage prediction. The ultimate goal of VaccineDesigner is to support streamlined, end-to-end analysis that enables the rational selection and assembly of immunogenic epitopes into optimized multi-epitope constructs with strong protective potential.

## 2. Materials and Methods

VaccineDesigner supports epitope prediction for B-cell and T-cell lymphocytes, as crucial indicators of protein antigenicity [40]. The architectural components are split into two discrete modules (Figure 1). Module A (epitope prediction) includes a set of tools for B-cell, CTL, and HTL epitope prediction, coupled with antigenicity, toxicity, and allergenicity assessment for the candidate epitopes of all cell types. Module B (multi-epitope vaccine construction) receives the individual prediction outputs of Module A and builds a high-quality multi-epitope vaccine library that includes relevant vaccine components and quality-filtering tools. VaccineDesigner applies the default parameter configuration of each tool in modules A and B, yet the parameters and thresholds of each tool are configurable according to user preferences.

The framework was designed to serialize and execute scientific tools via local installation, enabling the automation of well-established multi-epitope vaccine design workflows. The front end was developed using R Shiny, while R 4.2.2 and Python 3.10.5 scripts handle data analysis and tool execution. The two core modules are seamlessly integrated into a backend workflow that automates data flow, executes epitope prediction and evaluation tools sequentially, and ensures that intermediate outputs are properly formatted and passed between components without user intervention. Users can either analyze results directly or upload multiple epitope prediction files into the VaccineDesigner module to generate and evaluate final multi-epitope vaccine sequences. This integrated architecture enables complete, end-to-end analyses through a user-friendly graphical interface without requiring any programming expertise.

### 2.1. Epitope Prediction and Evaluation

#### 2.1.1. B-Cell Epitope Prediction

B-cell epitopes are predicted using BepiPred 3.0. BepiPred 3.0 systematically scans protein sequences and assesses key physicochemical properties and sequence patterns to identify regions that can induce robust antibody responses [19]. Based on the amino acid scores, VaccineDesigner generates B-cell epitopes, adhering to user-defined epitope length parameters. Two alternative analyses can be applied. The first involves the formation of B-cell epitopes in high-scoring regions. The lower length limit and the maximum number of amino acids between the predicted epitopes can be modified, enabling the combination of individual epitopes into larger immunogenic fragments. The second option pertains to B-cell epitope generation. Users can define a fixed-length scanning window to assess the scores of all amino acids. Eligible epitopes are set as continuous amino acid regions with a user-defined length, wherein each position must fulfill certain filtering criteria.

#### 2.1.2. CTL Epitope Prediction

VaccineDesigner performs CTL epitope prediction using NetMHCpan 4.1 [25] and the IEDB Consensus Method [41]. NetMHCpan implements an algorithm that identifies high-affinity regions for MHC class I alleles. Both methods predict strong and weak binder counts and export various pieces of information for the associated MHC alleles. The IEDB Consensus method combines the results of multiple prediction methods into a single consensus outcome, providing a more reliable assessment, as prediction algorithms may produce varying results due to differences in underlying models and assumptions. The final ranking of CTL epitopes is determined by the number of alleles interacting with each epitope.

#### 2.1.3. HTL Epitope Prediction

HTL epitope prediction relies on the NetMHCIIpan 4.0 framework. NetMHCIIpan 4.0 identifies regions of strong binding affinity for MHC class II molecules [25]. The results, akin to CTL epitopes, include strong and weak binder counts, along with information related to the MHC class II alleles. HTL epitope prediction can be performed by the IEDB Consensus method for class II binding prediction, employing the same framework as for the class I module [41]. The ranking of the HTL epitopes is determined by the number of interacting alleles, similar to the approach used for CTL epitopes.

#### 2.1.4. Epitope Evaluation

Given a set of candidate B-cell, CTL, and HTL epitopes, VaccineDesigner applies various quality metrics to filter out epitopes with suboptimal antigenicity and safety profiles, adhering to user-defined thresholds. Vaxign-ML [29] is used to comparatively assess the level of antigenicity for each epitope, prioritizing those that are highly immunogenic. To cope with potential safety issues, ToxinPred2 [32] is integrated into Module A to identify toxins within epitope sequences, and AlgPred 2.0 [31] to minimize the risk of allergic reactions. The epitopes that meet high-quality criteria for antigenicity, non-toxicity, and non-allergenicity are retained. CTL and HTL epitopes are further prioritized based on their binding affinity to multiple MHC alleles, while B-cell epitopes are selected based on high scoring and sequence continuity. This multi-layered filtering ensures the selection of safe, broadly immunogenic epitopes for vaccine construct generation.

### 2.2. Multi-Epitope Vaccine Sequences

The synthesis of multi-epitope constructs is considered a favorable approach to mitigate junctional immunogenicity and efficient epitope separation and presentation to T-cell and B-cell receptors 14. VaccineDesigner expands the analysis of candidate epitopes to craft multi-epitope vaccine sequences. Using the most prominent epitopes, VaccineDesigner fuses larger constructs by merging individual epitopes with appropriate linker sequences (default: GPGPG/HTL, AAY/CTL, KK/B-cell) [6,14,42,43,44,45,46] in a two-step procedure as described below.

#### 2.2.1. Vaccine Sequence Construction

The multi-epitope vaccine construct is built by a variable, user-defined number of B-cell, CTL, and HTL epitopes, as well as linker sequences between the epitopes and the N-terminus sequence. A variety of protein adjuvants are available for the synthesis of multi-epitope vaccine sequences [47,48,49,50]. In addition, the order of epitope combination can be defined in arrangements of B-cell, CTL, and HTL epitopes, exemplified as B-C-H (B-cell–CTL–HTL). VaccineDesigner generates candidate vaccine sequences that form a versatile library of epitope sequence combinations. Researchers can then explore diverse epitope combinations containing up to five epitopes from each category (B cells, CTLs, and HTLs).

#### 2.2.2. Vaccine Candidate Sequence Evaluation and Selection

In the final step, the candidate vaccine sequence library undergoes rigorous evaluation. The algorithm assesses each sequence against predefined filters, including Vaxign-ML for antigenicity, ToxinPred2 for toxin prediction, Algpred 2.0 for allergenicity, and ProtParam for physicochemical sequence analysis and stability [27]. User-defined parameters, such as the desired number of vaccine sequences, are used to track of the number of candidates that meet all quality criteria. The algorithm exports the final multi-epitope sequence constructs once the preferred number of candidates is reached. This dynamic approach ensures the selection of vaccine sequences that meet strict requirements for immunogenicity, safety, and biochemical attributes, offering the most promising candidates for further development and validation.

The candidate multi-epitope library is short-listed based on the ranking of the antigenicity, allergenicity, stability, and physicochemical properties of each construct. In addition, each candidate multi-epitope vaccine is scored by the deep neural network algorithm employed by DeepVacPred [51]. The analysis exports the list of candidate multi-epitope sequences once the user-defined maximum number of candidates is reached. The final candidate sequences are ranked based on the weighted sum of individual antigenicity, toxicity, allergenicity, stability, and the DeepVacPred rankings as follows: Let S be the vector of scores for each parameter, with $S_{i}\in R$ and $R\left(S\right)$ the corresponding rankings with $R\left(S_{i}\right)\in Z$ and $R\left(S\right)$ being a permutation of rankings, such that higher scores receive lower positions. In addition, let W be a vector of user-defined weights with $W_{i}\in R$. The best-scoring multi-epitope construct is the one that yields the smallest possible sum of weighted rankings, across the n ranking criteria i.e.,(1)$$min\left(\overset{5}{\underset{i=1}{\sum }}W_{i}\times R\left(S_{i}\right)\right)$$

The newly generated sequences can be re-analyzed to verify the favorable properties of individual epitopes in the whole construct and further validated for population coverage, molecular mimicry, and the presence of putative proteasome cleavages. Population coverage is a highly recommended analysis for human vaccine design and is implemented by the IEDB Population Coverage [36] algorithm. To investigate molecular mimicry, VaccineDesigner conducts protein similarity searches against SWISS-PROT [52] or metagenomic proteins (env_nr) using the BLASTp [53] algorithm. The analysis estimates whether a vaccine sequence shares common domains with a known host protein, thereby ensuring its antigenicity and minimizing the risk of autoimmune reactions. Furthermore, proteasome cleavages can be identified in the protein sequences using NetChop 3.1 [54]. The results are provided in both graphical and tabular formats.

## 3. Results

### 3.1. Usage Scenario

To demonstrate the functionalities of the VaccineDesigner workflow, we implemented an example usage scenario on the Atl (UniProt: P0C5Z8) and the IsdA (UniProt: Q7A152) proteins of Staphylococcus aureus. Atl is a bifunctional autolysin, playing a crucial role in the physiological processes, virulence, and pathogenesis of S. aureus. Autolysins are promising vaccine targets as they combine favorable features such as surface exposure, immunogenicity, and increased conservation among different strains. Iron-regulated surface determinant protein A (IsdA) is a virulence factor that is involved in the acquisition of iron, bacterial growth, and adherence to host tissues. IsdA combines desired vaccine properties, including surface exposure on the cell wall of S. aureus and induction of immune responses in infected individuals. IsdA, together with other cell wall-anchored surface proteins, has been reported to induce significant protective immunity in murine models [55].

In the usage scenario, 123 B-cell epitopes were predicted on the Atl and 136 on the IsdA protein using a fixed sequence length of 10 amino acids. Sixteen of these epitopes were of high quality based on their quantified toxicity, allergenicity, and antigenicity measures. A total of 96 and 18 CTL epitopes were identified in Atl and IsdA, respectively. These were predicted as strong or weak binders to MHC class I alleles, including HLA-A*01:01, HLA-A*01:02, HLA-A*01:03, HLA-A*01:04, HLA-A*01:06, HLA-A*01:07, HLA-A*01:08, and HLA-A*01:09. The predicted epitopes underwent toxicity, allergenicity, and antigenicity filtering, resulting in thirteen candidates. A total of 193 and 50 HTL epitopes were considered strong or weak binders to MHC class II alleles for the Atl and the IsdA protein, respectively, including HLA-DRB1*01:01, HLA-DBR1*01:02, HLA-DBR1*01:03, HLA-DBR1*01:04, HLA-DBR1*01:05, HLA-DBR1*01:06, HLA-DBR1*01:07, and HLA-DBR1*01:08. Allergenic and toxic epitopes accounted for a significant number of epitopes, resulting in 23 high-quality HTL epitopes for both proteins.

The list of the top-scoring epitopes of each category (B cells, CTLs, and HTLs) was further narrowed down to four to reduce the computational cost. The candidates were combined in the C-H-B order (CTLs, HTLs, and B cells). Two epitopes of each cell type were fused in the final constructs, resulting in 1728 candidate vaccine sequences. Figure 2a presents the alignments of the newly generated multi-epitope vaccine sequences. Figure 2b contains a ranking list of the first five multi-epitope sequences that passed all thresholds based on the level of antigenicity, toxicity, and related properties.

The final ranking of the high-quality, multi-epitope sequences (combined ranking) is defined by the collective sum of individual rankings. The best-scoring sequence (Seq2) including linker sequences is shown in Figure 3. Figure 3A shows the outcome of the proteasome cleavage analysis, in which the predicted cleavages are plotted for all the detected epitopes. Figure 3B depicts the multi-epitope sequence and its newly predicted B-cell, CTL, and HTL epitope regions after re-evaluating the synthesized construct in the multi-epitope vaccine construction module.

### 3.2. Evaluation Against Experimentally Validated Epitopes

To assess the competence of VaccineDesigner, we sought to compare the predictions of linear B-cell, cytotoxic T-cell (MHC I binding), and helper T-cell (MHC II binding) epitopes of the human herpesvirus (HSV-1) envelope glycoprotein D (UniProt ID: Q69091) against experimentally validated sequences provided by the IEDB database. Specifically, for each category, we retrieved experimental epitope data using IEDB search filters detailed in the Appendix A and then used the Immunome Browser [56] to visualize the locations of validated epitopes along the protein sequence. We subsequently analyzed the predictions from VaccineDesigner and compared their alignment against the experimentally verified epitopes.

Testing procedure: The configuration parameters of each tool were set using the integrated graphical environment of VaccineDesigner. To quantify the level of antigenicity, Vaxign-ML was configured for viruses 21 with a protegenicity threshold set to 0.3 for each epitope. Since viruses do not produce toxins and their pathogenicity arises from their infection and replication cycle [57], ToxinPred2 was applied with a higher threshold (0.8), allowing for a less stringent prediction. Similarly, AlgPred 2.0 was set to a higher threshold (0.45) to reduce strictness. Since the experimentally validated epitopes lack full toxicity and allergenicity data, these high-threshold filters were primarily used to ensure a comprehensive evaluation of VaccineDesigner’s pipeline, with a focus on mapping the predicted epitopes for validation. BepiPred 3.0 was employed to predict B-cell epitopes of variable lengths, with the threshold set to 0.17, closely aligned with the default threshold of 0.1512, to ensure a stricter selection of high-quality epitopes within the protein. The parameter Subthreshold Amino Acid Inclusion Count was set to 2 to enable the synthesis of larger continuous epitope regions that were separated by two single lower-score amino acids (Secondary threshold: 0.12). CTL and HTL epitopes were predicted on the most common HLA alleles identified from assay information derived from the IEDB database (MHC I molecules: HLA-A*02:01, HLA-A*02:03, HLA-A*02:06, and MHC II molecules: HLA-DQA1*01:01/DQB1*03:35, HLA-DRB1*04:04, DRB1*01:03, DRB5*01:01, DRB1*01:01, DRB1*04:01, DRB1*13:01). VaccineDesigner was set to run NetMHCpan-4.1 and netMHCIIpan-4.0 with peptide lengths of 9 and 15, respectively [58,59].

Results: VaccineDesigner exported four variable-length B-cell epitopes, ranging between 12 and 27 amino acids. All predictions overlap with, on average, eighteen experimentally validated epitopes. Figure 4A shows a significant overlap between the predicted B-cell epitopes and the experimentally validated ones. From the nineteen predicted CTL epitopes, we identified eleven that uniquely bind to all three MHC I molecules, demonstrating significant overlap with the experimentally validated epitopes. For the 75 predicted HTL epitopes, we focused on those that interacted with more than one MHC II molecule, irrespective of their binding strength, which again showed substantial overlap with validated epitopes. Figure 4B and 4C show the residual overlaps between the predicted MHC-I and MHC-II epitopes and the validated nine MHC-I and 43 MHC-II epitopes from IEDB (IEDB v.3.10.0). The results, the parameters of VaccineDesigner, and IEDB configurations are provided in the Appendix A.

Comments: It is important to note that the selection of MHC molecules typically relies on the available literature and existing database information, which may not encompass all possible MHC interactions. Consequently, the full range of results for MHC-I and MHC-II interactions may not be captured. Further comparative analyses are necessary to understand how different parameter configurations can affect overall performance, as exhaustive manual testing of these configurations was not possible due to resource limitations. By enabling transparent execution of the pipelined processes, VaccineDesigner significantly reduces execution time, which is crucial when thoroughly examining a candidate protein under varying configurations. Furthermore, despite the above concerns, it is noteworthy that the protein analyzed contains a relatively small number of validated epitopes compared to more extensively studied immunogenic proteins, and VaccineDesigner effectively captured the most critical positions.

### 3.3. Comparison with Other RV Pipelines

VaccineDesigner incorporates unique features within established reverse vaccinology (RV) workflows, aiming to streamline multi-epitope vaccine construction within a user-friendly environment. Table 1 outlines essential functions of popular RV tools, detailing aspects such as web accessibility, epitope type support, non-human species compatibility, pathogen protein selection, candidate evaluation, and source code availability. Unlike other RV tools, VaccineDesigner enables multi-epitope vaccine construction directly through a Web interface. OptiVax [60], while supporting multi-epitope vaccine design, is limited to T-cell epitopes and is not accessible through a Web or standalone graphical user interface. In addition, among single-epitope predictors, ReVac [61] and VacSOL [62] are only accessible command-line, while IEDB-AR [36] does not support streamlined analyses. VaccineDesigner is the only web-based application that combines streamlined analyses with B-cell and T-cell epitope prediction. Overall, VaccineDesigner uniquely supports automated multi-epitope construct generation, re-analysis, and prioritization using customizable thresholds and a weighted scoring system, capabilities not available with existing tools. These features, coupled with comprehensive filtering and evaluation criteria for the multi-epitope vaccine candidates, make VaccineDesigner a versatile tool for researchers working on vaccines for both human and non-human applications.

## 4. Discussion

VaccineDesigner was developed to build protective multi-epitope constructs using in silico B-cell, CTL, and HTL epitope prediction methods coupled with toxicity, allergenicity, and antigenicity assessment. Candidate multi-epitope vaccines are further examined to estimate population coverage, molecular mimicry, and proteasome cleavage to maximize protective immune responses. Technically, VaccineDesigner aims to address the reported underutilization of automated workflows in reverse vaccinology by enabling pipelined analyses through a user-friendly graphical environment that includes preconfigured software tools and effortless I/O data flow management [64].

VaccineDesigner extends RV frameworks by introducing several technical and functional advancements: (1) It implements a stepwise procedure that is initiated with a list of candidate antigenic proteins and progressively leads to the most likely effective multi-epitope vaccines. (2) VaccineDesigner leverages the collective strengths of each tool and combines their outputs in unified tabular and graphical reports. (3) A weighted scoring scheme is applied to prioritize multi-epitope constructs based on individual properties. (4) It ensures time-efficient and effortless application due to the seamless data flow between cascading tasks implemented by individual tools. (5) VaccineDesigner operates independently of local installations and operating system restrictions and specifications.

Considering the architectural components, VaccineDesigner is built on a foundation of well-validated methodologies recognized and trusted within the scientific community. The selection of the tools was primarily based on benchmarking studies [29,64], citation-based recognition by the scientific community, and on the technical constraints that are related to the programmatic access and availability of the source code. A potential improvement enabled by the modular architecture of VaccineDesigner is to combine evidence from multiple tools performing the same analysis. However, the lack of interoperable tools presents additional technical challenges as it is hard to standardize inputs, parameters, and outputs, and, most importantly, to provide sophisticated and more effective consensus algorithms. Considering the significant number of informative properties that are produced for each candidate protein, a reasonable enhancement would be to mine discriminative properties through machine learning models to predict peptides that induce protective immune responses. Still, the impact of these models is limited by the inaccuracies of the positive and negative training data as well as the lack of a definitive list of analyses that all RV pipelines should include [64].

Another important limitation is that VaccineDesigner does not currently evaluate the 3D structure or folding of the final multi-epitope constructs. While sequence-based evaluation promotes epitope quality and immunogenicity potential, it cannot predict how the combined epitopes will fold or whether the final construct will be suitable for recombinant expression. Misfolding can affect both expression yield and immune accessibility of the epitopes. To address these limitations, our platform provides links to supplementary validation tools that users can independently access for expression-related assessments: Protein-Sol [65] solubility prediction, OPTIMIZER [66] for codon usage optimization, RNAfold [67] for mRNA secondary structure analysis, and ColabFold [68] for 3D structure prediction. While these analyses provide valuable preliminary assessments, further structural validation through molecular docking with immune receptors and molecular dynamics simulations can offer deeper insights into construct viability, though these approaches require specialized expertise and computational resources beyond the scope of our streamlined screening platform.

It should be noted that VaccineDesigner does not address the initial protein selection stage. Practically, the starting point of the pipeline is a set of proteins that have already been selected as potentially immunogenic. To incorporate this analysis several challenges must be addressed, including the selection of proteins based on the subcellular localization, the exclusion of homologs to avoid autoimmune reactions in the host, and the use of pangenomes to select core genome components that reduce the selective pressure [69]. Moving to more holistic approaches, VaccineDesigner will pursue the integration of 3D modeling algorithms to predict the folded 3D structure of the multi-epitope vaccine constructs and to identify exposed peptides that are more likely to interact with immune cells. Alphafold 3 represents a transformative advancement in structural biology, with the potential to accelerate precision vaccine design by predicting structures of DNA, RNA, ligands, and other key biomolecules [70].

## 5. Conclusions

In the evolving landscape of infectious diseases, multi-epitope vaccine design has emerged as a pivotal strategy to identify antigenic determinants in pathogenic proteins while also reducing the risk of adverse reactions and insufficient coverage against diverse pathogenic strains. VaccineDesigner is a reverse vaccinology framework that combines an automated procedure for the prediction and assessment of protective multi-epitope constructs within a fully customizable, graphical web-based environment. As our insights into host–pathogen interactions advance, we anticipate that VaccineDesigner will provide a valuable tool for rationalized knowledge synthesis and will contribute to real-world applications as a versatile framework for developing safer and more effective vaccines.

While VaccineDesigner offers a comprehensive framework for multi-epitope vaccine design, it currently does not perform the initial antigen selection from pathogen proteomes. Future development will focus on integrating protein prioritization modules based on subcellular localization, virulence, and homology filtering. Additionally, we aim to incorporate recombinant expression assessment capabilities alongside 3D structure modeling for epitope accessibility and to improve consensus scoring methods using machine-learning-based ensemble approaches.

## Figures and Tables

**Figure 1 biology-14-01019-f001:**
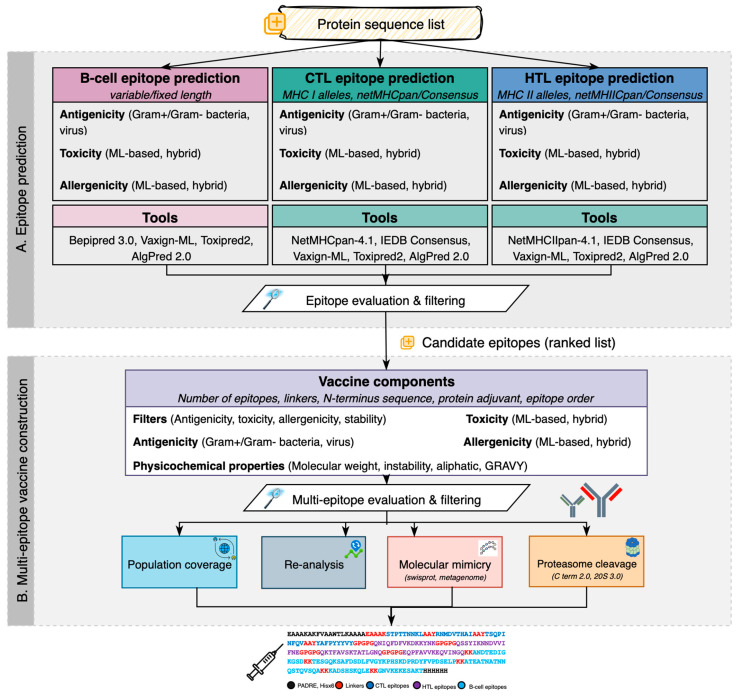
The main functional components of the streamlined process implemented by VaccineDesigner. (**A**) The epitope prediction module receives a FASTA protein file, performs epitope prediction (B cells, CTLs, HTLs), and concludes with evaluating and filtering epitope sequences based on their antigenicity, toxicity, and physicochemical properties. (**B**) The multi-epitope vaccine construction module receives the tabular-formatted results containing the candidate epitopes and generates multi-epitope vaccine sequences based on user-defined thresholds. To assess the quality of the candidate vaccine sequences, users can investigate the population coverage, molecular mimicry, and proteasome cleavages as well as re-analyze the construct using the supported single-epitope prediction modules.

**Figure 2 biology-14-01019-f002:**
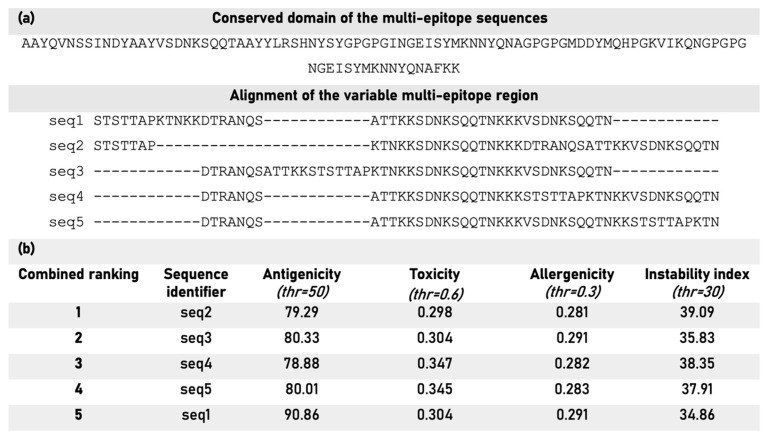
(**a**) A conserved domain of 98 amino acids is present in the five high-scoring multi-epitope sequences (top), along with the aligned variable region of 46 amino acid residues. (**b**) Combined ranking of the five multi-epitope sequence candidates along with their predicted antigenicity, toxicity, allergenicity, and instability index.

**Figure 3 biology-14-01019-f003:**
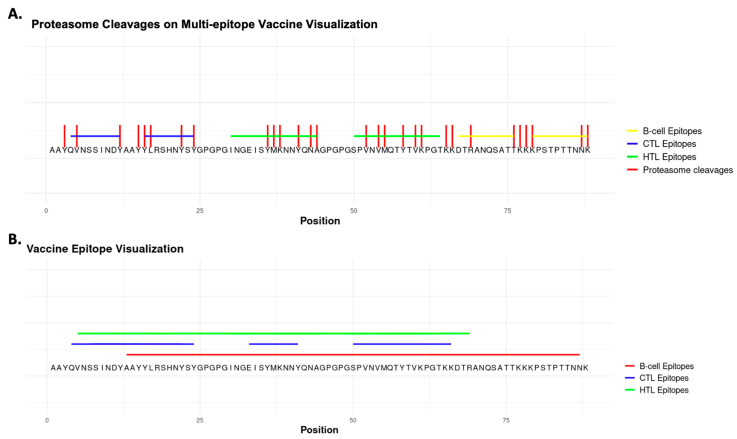
Visualization of multi-epitope vaccine based on the proteasome cleavage functionality (**A**), and multi-epitope sequence re-evaluation with the epitope prediction module (**B**).

**Figure 4 biology-14-01019-f004:**
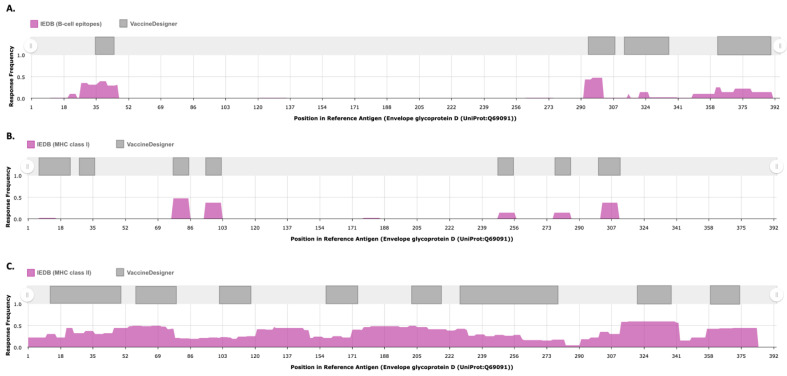
Overlapping residual loci between the top-scoring B-cell (**A**), CTL (**B**), and HTL (**C**) predictions and the response frequency curves generated by Immunome Browser 3.0^56^ using IEDB v.3.10.0.

**Table 1 biology-14-01019-t001:** Main features of RV pipelines.

Name/Tool	Web Interface	Streamlined Process (Yes/No)	Epitope Types (BP/TP/BP + TP)	Non-Human Species (Yes/No)	Multi-Epitope Construction (Yes/No)	Pathogen Protein Selection (Yes/No)	Vaccine Candidate Evaluation (*)	Source Code Availability (Yes/No)
NERVE [63]	No	Yes	No	No	No	Yes	SL, AP, CR	Yes
iVAX [34] **	Yes	Yes	TP	Yes	No	Yes	I, C, CR, A	No
OptiVax [60]	No	Yes	TP	Yes	Yes	Yes	I, PC, SP	Yes
ReVac [61]	No	Yes	BP + TP	Yes	No	Yes	A, C, SL, CR	Yes
IEDB-AR [36]	Yes	No	BP + TP	Yes	No	No	No	Yes
Vacceed [35]	No	Yes	TP	Yes	No	Yes	SL, SP, TH, T	Yes
VacSOL [62]	No	Yes	BP + TP	No	No	Yes	SL, V, TH, A, B + T	Yes
Vaxign2 [33]	Yes	Yes	TP	No	No	Yes	SL, TH, AP, B + T	Yes
VaccineDesigner	Yes	Yes	BP + TP	Yes	Yes	No	B + T, A, S, CR, PC	Yes

* SL: subcellular localization, AP: adhesin probability, I: immunogenicity, C: conservation, PC: population coverage, A: antigenicity, CR: cross-reactivity, S: signal peptide, TH: transmembrane helices, V: virulence, BP: B-cell prediction, TP: T-cell prediction, BP + TP: B-cell and T-cell prediction, B + T: presence of B-cell and T-cell epitopes, T: presence of T-cell epitopes, S: stability; ** iVAX is commercial software.

## Data Availability

VaccineDesigner is an open-source tool freely available under academic license at https://github.com/BiolApps/VaccineDesigner. Project name: VaccineDesigner. Project home page: http://bioinformatics.med.auth.gr/VaccineDesigner. Operating system(s): platform-independent. Programming language: R, Python. Other requirements: web browser. License: academic free license. Any restrictions to use by non-academics: license needed.

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
