# Peer review of "VaccineDesigner: A Web-Based Tool for Streamlined Multi-Epitope Vaccine Design"

_biology, 2025, doi:10.3390/biology14081019_

Round 1

Reviewer 1 Report

Comments and Suggestions for Authors

The manuscript "VaccineDesigner: A Web-based Tool for Streamlined Multi-epitope Vaccine Design" reports the development of a web-based tool for designing multi-epitope vaccines. This study can provide key insights into the development of new vaccines using AI in the future. The study is well-designed and scientifically sound. The manuscript can be accepted after minor corrections. 

  1. Introduction: Provide insights into the current study with its novelty and scope.
  2. How is the developed vaccine designer web tool advantageous over previously reported tools, and what kind of edge does it provide that is not possible in previous tools?
  3. What are the selection criteria for the selected three epitomes (B-cell, CTL, and HTL epitopes)? 
  4. Conclusion: Add limitations and future scope of the current findings.
  5. The manuscript needs to be revised for minor typos and grammatical errors. 

Author Response

Comment 1:The manuscript "VaccineDesigner: A Web-based Tool for Streamlined Multi-epitope Vaccine Design" reports the development of a web-based tool for designing multi-epitope vaccines. This study can provide key insights into the development of new vaccines using AI in the future. The study is well-designed and scientifically sound. The manuscript can be accepted after minor corrections.

Response 1: We are grateful for the positive feedback and thoughtful evaluation of our manuscript titled "VaccineDesigner: A Web-based Tool for Streamlined Multi-epitope Vaccine Design." We are pleased that the reviewer found the study scientifically sound, well-designed, and a valuable contribution to the field of vaccine development.

Comment 2:Introduction: Provide insights into the current study with its novelty and scope.

Response 2: We thank the reviewer for the valuable suggestion. In response, we have revised the Introduction to better highlight the novelty and scope of the current study. We changed the last paragraph of Introduction section (lines 103-112 of initial manuscript) with the following text (lines 106-118):

To address these limitations, we introduce VaccineDesigner, a novel, open-access, Web-based platform that provides a fully integrated, end-to-end workflow for rational multi-epitope vaccine design. The platform supports comprehensive epitope prediction for both B-cell and T-cell responses, facilitating vaccine development for a broad range of human and non-human pathogens. Rather than requiring manual integration of heterogeneous and distributed tools, VaccineDesigner executes cascading tasks seamlessly within a unified, customizable graphical interface.

Key functionalities include multi-epitope vaccine generation, candidate prioritization, population coverage estimation, molecular mimicry analysis, and proteasome cleavage prediction. The ultimate goal of VaccineDesigner is to support streamlined, end-to-end analysis that enables the rational selection and assembly of immunogenic epitopes into optimized multi-epitope constructs with strong protective potential.

Comment 3:How is the developed vaccine designer web tool advantageous over previously reported tools, and what kind of edge does it provide that is not possible in previous tools?

Response 3: We thank the reviewer for the constructive question. VaccineDesigner offers a clear advantage over existing tools by providing a fully integrated, Web-based platform that supports B-cell, CTL, and HTL epitope prediction, automated multi-epitope vaccine construction, and comprehensive evaluation within a single interface. Unlike existing tools that handle isolated steps or require manual processing, VaccineDesigner streamlines the entire workflow and enables users to generate, rank, and re-analyze candidate vaccine constructs in a user-friendly and reproducible manner. These features are summarized and contrasted with other tools in Table 1. To clarify this advantage, we have added the following sentences (lines 401-407) to “3.3 Comparison with other RV pipelines” paragraph.

Overall, VaccineDesigner uniquely supports automated multi-epitope construct generation, re-analysis, and prioritization using customizable thresholds and a weighted scoring system, capabilities not available in existing tools. These features, coupled with comprehensive filtering and evaluation criteria for the multi-epitope vaccine candidates, make VaccineDesigner a versatile tool for researchers working on vaccines for both human and non-human applications.”

Comment 4:What are the selection criteria for the selected three epitomes (B-cell, CTL, and HTL epitopes)?

Response 4: We thank the reviewer for the question. The selected B-cell, CTL, and HTL epitopes were filtered based on their antigenicity (Vaxign-ML), non-toxicity (ToxinPred2), and non-allergenicity (AlgPred 2.0). CTL and HTL epitopes were further prioritized based on binding affinity to multiple MHC alleles (NetMHCpan 4.1, NetMHCIIpan 4.0), while B-cell epitopes were selected using BepiPred 3.0 based on sequence scoring and continuity. These criteria ensured the selection of high-quality, safe, and broadly immunogenic epitopes. To further clarify the criteria for the selection of the epitopes, we revised the last paragraph (lines 212-217) of 2.1.4 Epitope Evaluation section.

“The epitopes that meet high-quality criteria for antigenicity, non-toxicity, and non-allergenicity are retained. CTL and HTL epitopes are further prioritized based on their binding affinity to multiple MHC alleles, while B-cell epitopes are selected based on high scoring and sequence continuity. This multi-layered filtering ensures the selection of safe, broadly immunogenic epitopes for vaccine construct generation.”

Comment 5: Conclusion: Add limitations and future scope of the current findings. 

Response 5: We thank the reviewer for the helpful suggestion. In response, we have revised the Conclusion section to briefly address the limitations of the current study and highlight the future scope. The following sentences (lines 494-500) have been added to the end of the “Conclusion” section:

“While VaccineDesigner offers a comprehensive framework for multi-epitope vaccine design, it currently does not perform the initial antigen selection from pathogen proteomes. Future development will focus on integrating protein prioritization modules based on subcellular localization, virulence, and homology filtering. Additionally, we aim to incorporate 3D structure modeling for epitope accessibility and improve consensus scoring methods using machine learning-based ensemble approaches. Additionally, we aim to incorporate recombinant expression assessment capabilities alongside 3D structure modeling for epitope accessibility and to improve consensus scoring methods using machine learning-based ensemble approaches.”

Comment 6: The manuscript needs to be revised for minor typos and grammatical errors.

Response 6: We appreciate the reviewer’s suggestion. We have carefully revised the manuscript and made the necessary corrections to address all minor typos and grammatical errors.

Reviewer 2 Report

Comments and Suggestions for Authors

This is a thoroughly biocomputational-based paper that is aimed to offer a streamlined integrated tool for prokaryotic multi-epitope vaccine design for a given pathogen as a practical realization of Reverse Vaccinology  For that, the authors have applied a good assortment of available well-validated biocomputational platforms and data. This tool combines B-cell, CTL, and HT  epitope predictions, potentially from different immunogens.

            The paper is well written, presented and understandable.

            Table 1 summarizes main features of VaccineDesigner in comparison with other Reverse Vaccinology pipelines.

            This is just a biocomputational paper not accompanied by any experimental validation. The authors are aware of some limitations of their approach. By excluding potential toxicity and molecular mimicry issues, it is expected that this tool will help to design and develop safer and more effective vaccines.

Author Response

Reviewer 2

Comment 1:This is a thoroughly biocomputational-based paper that is aimed to offer a streamlined integrated tool for prokaryotic multi-epitope vaccine design for a given pathogen as a practical realization of Reverse Vaccinology  For that, the authors have applied a good assortment of available well-validated biocomputational platforms and data. This tool combines B-cell, CTL, and HT  epitope predictions, potentially from different immunogens.

The paper is well written, presented and understandable.

Table 1 summarizes main features of VaccineDesigner in comparison with other Reverse Vaccinology pipelines.

 This is just a biocomputational paper not accompanied by any experimental validation. The authors are aware of some limitations of their approach. By excluding potential toxicity and molecular mimicry issues, it is expected that this tool will help to design and develop safer and more effective vaccines.

Response 1: We sincerely thank the reviewer for the encouraging and thoughtful comments. We appreciate the recognition of our effort to integrate existing state-of-the-art and well-established computational tools into a streamlined framework for multi-epitope vaccine design. As noted, this work represents a computational pipeline without experimental validation; however, we believe it lays the groundwork for future experimental studies by enabling rapid, reproducible, and systematic vaccine candidate prioritization. We also acknowledge the reviewer’s point regarding the tool’s focus on minimizing potential toxicity and molecular mimicry, which are crucial for enhancing vaccine safety and efficacy.

Reviewer 3 Report

Comments and Suggestions for Authors

In this manuscript, the authors propose a novel, comprehensive web-based framework for designing multi-epitope vaccines. This framework streamlines the construction of protective epitope-based vaccines by seamlessly integrating computational methods for B-cell, CTL, and HTL epitope prediction.

The idea is interesting, but the Journal is not the best option for a thorough revision. From my perspective, as a non-expert in programming and with limited experience in the field, the manuscript requires more details about the construction of this integrated computational method. Additionally, the sole prediction of a multi-epitope protein (vaccine) does not guarantee proper folding for correct recombinant expression. This is a significant limitation of this method. Unfortunately, I do not see the added value or advantage of this method.

Author Response

Comment 1:The idea is interesting, but the Journal is not the best option for a thorough revision.

Response 1: We sincerely thank the reviewer for the thoughtful comment. Our manuscript was submitted to "Biology" Journal for consideration under the "Bioinformatics" section following the editorial board’s recommendation that expressed a strong interest in our work. We respectfully believe that the manuscript aligns well with the stated scope of the Bioinformatics section of the journal. Our work presents a novel, web-based bioinformatics platform which integrates state-of-the-art computational methods for the prediction and evaluation of multi-epitope vaccine candidates. The platform is developed to streamline, simplify, and democratize the implementation of reverse vaccinology applications by providing a publicly accessible, automated workflow that integrates a comprehensive suite of relevant and technically heterogeneous bioinformatics tools. The development and integration of such computational pipelines directly support the section’s objective to publish new bioinformatics tools and methods that enable discoveries across biology-relevant fields. Moreover, our study focuses on vaccine design for both human and non-human pathogens, a topic of high relevance to microbiology, immunology, genomics, and infectious disease biology. Therefore, with the support of the editorial board of the Bioinformatics section, we believe that VaccineDesigner represents a valuable resource for biomedical researchers, promoting reproducibility, accessibility, and efficiency in epitope-based vaccine research.

Comment 2: From my perspective, as a non-expert in programming and with limited experience in the field, the manuscript requires more details about the construction of this integrated computational method.

Response 2: We thank the reviewer for the helpful comment. To address the concern regarding the clarity of the integrated computational method, we have revised the Materials and Methods section (specifically in the description of the VaccineDesigner architecture) to provide a clearer explanation of how the tools are linked. We now clarify that the two main modules—epitope prediction and multi-epitope vaccine construction—are fully integrated in a backend workflow that automates data flow, sequential execution, and output formatting. This ensures that users can carry out the full analysis without requiring any programming skills. These additions aim to improve accessibility for readers with limited technical background. The following text has been added in the “Material and Methods” section (lines 130-140):

The framework was designed to serialize and execute scientific tools via local installation, enabling the automation of well-established multi-epitope vaccine design workflows. The front end was developed using R Shiny, while R and Python scripts handle data analysis and tool execution. The two core modules are seamlessly integrated into a backend workflow that automates data flow, executes epitope prediction and evaluation tools sequentially, and ensures that intermediate outputs are properly formatted and passed between components without user intervention. Users can either analyze results directly or upload multiple epitope prediction files into the VaccineDesigner module to generate and evaluate final multi-epitope vaccine sequences. This integrated architecture enables complete, end-to-end analyses through a user-friendly graphical interface—without requiring any programming expertise.”

Comment 3:Additionally, the sole prediction of a multi-epitope protein (vaccine) does not guarantee proper folding for correct recombinant expression. This is a significant limitation of this method.

Response 3:

We fully agree with the reviewer that the prediction of multi-epitope vaccine constructs does not guarantee correct folding or successful recombinant expression, and this represents a well-recognized limitation of in silico approaches.

The primary contribution of our work is to facilitate the computational screening and evaluation of peptide regions, with the aim of reducing experimental time and cost, rather than focusing on structural validation or production-level construct optimization.

The individual tools integrated in our framework are widely used for in silico vaccine candidate screening. However, when used independently, they produce incompatible results that cannot be easily integrated. Our pipeline addresses this by creating a sequential workflow where tools feed into each other to assess epitopic regions and construct multiepitope peptides.

The multiepitope construction module implements linker methodologies derived from fusion protein technology and emerging multiepitope vaccine studies. To support early evaluation, ProtParam is included to provide basic physicochemical profiling, even though it can’t predict folding or expression outcomes.

The pipeline generates multiepitope constructs with comprehensive assessments, but in silico recombinant expression and folding analyses provide descriptive outputs of the final constructs rather than filtering criteria for subsequent pipeline steps. Therefore, we decided to provide web-links of supplementary tools for potential interested users in the final results tab of the VaccineDesigner web-interface, to independently evaluate expression-related properties:

i)Protein-Sol (https://protein-sol.manchester.ac.uk/) - Protein solubility prediction for recombinant expression assessment.

ii)OPTIMIZER (http://genomes.urv.es/OPTIMIZER/) - Codon usage optimization to enhance expression levels in selected hosts.

iii)RNAfold (http://rna.tbi.univie.ac.at/cgi-bin/RNAWebSuite/RNAfold.cgi) - mRNA secondary structure analysis to evaluate ribosome binding site accessibility and translation efficiency.

iv)ColabFold (https://colab.research.google.com/github/sokrypton/ColabFold/blob/main/AlphaFold2.ipynb) - Protein structure prediction including secondary structure elements and 3D folding assessment for multiepitope constructs.

While these provide valuable preliminary assessments, advanced structural validation through molecular docking and dynamics simulations requires specialized expertise beyond our platform's scope.

Following the reviewer's comment regarding the limitations of in silico folding prediction, we have modified the manuscript to clearly state these constraints and emphasize the screening nature of our approach.

Added to the Discussion (lines 456–470):

“Another important limitation is that VaccineDesigner does not currently evaluate the 3D structure or folding of the final multi-epitope constructs. While sequence-based evaluation promotes epitope quality and immunogenicity potential, it cannot predict how the combined epitopes will fold or whether the final construct will be suitable for recombinant expression. Misfolding can affect both expression yield and immune accessibility of the epitopes. To address these limitations, our platform provides links to supplementary validation tools that users can independently access for expression-related assessments: Protein-Sol [65] for solubility prediction, OPTIMIZER [66]  for codon usage optimization, RNAfold [67] for mRNA secondary structure analysis, and ColabFold [68] for 3D structure prediction. While these analyses provide valuable preliminary assessments, further structural validation through molecular docking with immune receptors and molecular dynamics simulations can offer deeper insights into construct viability, though these approaches require specialized expertise and computational resources beyond the scope of our streamlined screening platform.”

We also take the reviewer’s comment into serious consideration and plan to investigate a possible in silico optimization of recombinant expression capabilities in the future.

Added to the Conclusion (lines 494–500):

“While VaccineDesigner offers a comprehensive framework for multi-epitope vaccine design, it currently does not perform the initial antigen selection from pathogen proteomes. Future development will focus on integrating protein prioritization modules based on subcellular localization, virulence, and homology filtering. Additionally, we aim to incorporate recombinant expression assessment capabilities alongside 3D structure modeling for epitope accessibility and to improve consensus scoring methods using machine learning-based ensemble approaches.”

Our tool serves as a screening platform for immunogenic regions and multiepitope combinations, consistent with in silico methods in drug discovery that reduce candidate pools and assess molecular features before experimental validation. Recognizing recombinant expression as a key challenge, we provide links to validation tools and emphasize the importance of experimental confirmation. We appreciate the reviewer’s insightful comment and hope our response fully addresses it.

Comment 4:Unfortunately, I do not see the added value or advantage of this method.

Response 4: We thank the reviewer for the thoughtful comment. We respectfully note that while existing reverse vaccinology tools often provide epitope prediction capabilities, they typically focus on isolated tasks and lack an integrated, end-to-end workflow for multi-epitope vaccine design.

VaccineDesigner’s added value lies in its unique ability to combine B-cell, CTL, and HTL epitope prediction with advanced filtering for safety and immunogenicity, along with automated construction and prioritization of multi-epitope constructs within a single, user-friendly Web-based platform. Additionally, it offers downstream analyses such as population coverage estimation, molecular mimicry detection, and proteasome cleavage prediction—capabilities that are not comprehensively addressed by other tools. Table 1 of the manuscript provides a detailed comparison between VaccineDesigner and other reverse vaccinology platforms. Moreover, Section 3.3 “Comparison with other RV pipelines” highlights the distinctive strengths of our methodology relative to existing approaches. Most importantly, VaccineDesigner automates widely used but time-consuming RV pipelines that typically require manual integration of heterogeneous and distributed tools, providing researchers and immunologists with a unified framework that streamlines analysis from epitope prediction to final construct evaluation.

Computational screening tools like VaccineDesigner serve as platforms for reducing candidate pools and evaluating molecular features prior to experimental validation. While recombinant expression and folding remain significant challenges that require specialized structural validation, VaccineDesigner provides essential preliminary screening by selecting high-quality, safe, and immunogenic epitopes, a necessary foundation for subsequent optimization steps.

The practical impact is significant: tasks that previously demanded expertise across multiple tools, manual data formatting, and complex integration workflows can now be executed seamlessly through a single web-based interface. This provides broader access to integrated vaccine design capabilities and reduces the time required to progress from protein input to ranked multiepitope constructs.

While computational screening cannot solve all challenges (as we acknowledged regarding structural validation), VaccineDesigner addresses the critical bottleneck of tool integration and workflow automation that currently limits widespread adoption of reverse vaccinology approaches.

Fundamentally, multi-epitope vaccines represent a promising strategy that, if successfully developed, could simultaneously elicit comprehensive B-cell and T-cell immune responses while avoiding potentially harmful pathogenic components—making computational frameworks essential for advancing research in this challenging but potentially transformative area of vaccine development.

The following lines were added in the revised manuscript:

Introduction (lines 106-118):

“To address these limitations, we introduce VaccineDesigner, a novel, open-access, Web-based platform that provides a fully integrated, end-to-end workflow for rational multi-epitope vaccine design. The platform supports comprehensive epitope prediction for both B-cell and T-cell responses, facilitating vaccine development for a broad range of human and non-human pathogens. Rather than requiring manual integration of heterogeneous and distributed tools, VaccineDesigner executes cascading tasks seamlessly within a unified, customizable graphical interface.

Key functionalities include multi-epitope vaccine generation, candidate prioritization, population coverage estimation, molecular mimicry analysis, and proteasome cleavage prediction. The ultimate goal of VaccineDesigner is to support streamlined, end-to-end analysis that enables the rational selection and assembly of immunogenic epitopes into optimized multi-epitope constructs with strong protective potential.”

3,3 Comparison with other RV tools (lines 401-407):

Overall, VaccineDesigner uniquely supports automated multi-epitope construct generation, re-analysis, and prioritization using customizable thresholds and a weighted scoring system, capabilities not available in existing tools. These features, coupled with comprehensive filtering and evaluation criteria for the multi-epitope vaccine candidates, make VaccineDesigner a versatile tool for researchers working on vaccines for both human and non-human applications.”

Discussion (lines 456-470):

Another important limitation is that VaccineDesigner does not currently evaluate the 3D structure or folding of the final multi-epitope constructs. While sequence-based evaluation promotes epitope quality and immunogenicity potential, it cannot predict how the combined epitopes will fold or whether the final construct will be suitable for recombinant expression. Misfolding can affect both expression yield and immune accessibility of the epitopes. To address these limitations, our platform provides links to supplementary validation tools that users can independently access for expression-related assessments: Protein-Sol [65] solubility prediction, OPTIMIZER [66] for codon usage optimization, RNAfold [67] or mRNA secondary structure analysis, and ColabFold [68] for 3D structure prediction. While these analyses provide valuable preliminary assessments, further structural validation through molecular docking with immune receptors and molecular dynamics simulations can offer deeper insights into construct viability, though these approaches require specialized expertise and computational resources beyond the scope of our streamlined screening platform.”
